# The Variations of Leaf δ^13^C and Its Response to Environmental Changes of Arbuscular and Ectomycorrhizal Plants Depend on Life Forms

**DOI:** 10.3390/plants11233236

**Published:** 2022-11-25

**Authors:** Shan Zhang, Mingli Yuan, Zhaoyong Shi, Shuang Yang, Mengge Zhang, Lirong Sun, Jiakai Gao, Xugang Wang

**Affiliations:** 1College of Agriculture, Henan University of Science and Technology, Luoyang 471023, China; 2Luoyang Key Laboratory of Symbiotic Microorganism and Green Development, Luoyang 471023, China; 3Henan Engineering Research Center of Human Settlements, Luoyang 471023, China; 4School of Agriculture and Animal Husbandry Engineering, Zhoukou Vocational and Technical College, Zhoukou 466000, China

**Keywords:** leaf δ^13^C, arbuscular mycorrhiza (AM), ectomycorrhiza (ECM), environment factors, life forms

## Abstract

Arbuscular mycorrhiza (AM) and ectomycorrhiza (ECM) are the two most common mycorrhizal types and are paid the most attention to, playing a vital common but differentiated function in terrestrial ecosystems. The leaf carbon isotope ratio (δ^13^C) is an important factor in understanding the relationship between plants and the environment. In this study, a new database was established on leaf δ^13^C between AM and ECM plants based on the published data set of leaf δ^13^C in China’s C_3_ terrestrial plants, which involved 1163 observations. The results showed that the differences in leaf δ^13^C between AM and ECM plants related closely to life forms. Leaf δ^13^C of ECM plants was higher than that of AM plants in trees, which was mainly led by the group of evergreen trees. The responses of leaf δ^13^C to environmental changes were varied between AM and ECM plants. Among the four life forms, leaf δ^13^C of ECM plants decreased more rapidly than that of AM plants, with an increase of longitude, except for deciduous trees. In terms of the sensitivity of leaf δ^13^C to temperature changes, AM plants were higher than ECM plants in the other three life forms, although there was no significant difference in evergreen trees. For the response to water conditions, the leaf δ^13^C of ECM plants was more sensitive than that of AM plants in all life forms, except evergreen and deciduous trees. This study laid a foundation for further understanding the role of mycorrhiza in the relationship between plants and the environment.

## 1. Introduction

Leaf carbon isotope discrimination (δ^13^C) plays an important role in our understanding of the relationship between plants and environment. It not only provides a series of climate and environmental information related to plant growth processes, but it can also indicate how plants interact with the environment, and respond [1,2,3,4]. Thus, the relationship between leaf δ^13^C and the environment has aroused interest [5,6,7,8].

There are many studies which have demonstrated that the characteristics of the plant itself can affect the leaf δ^13^C [9,10]. Study found significant differences in δ^13^C among different species of *R. natans*, which may be caused by stomatal limitation [4]. Li et al. [7] found the order of the averaged δ^13^C for plant life forms from most positive to most negative was subshrubs > herbs = shrubs > trees > subtrees after studying 2538 plants in China. He et al. [11] found that tree height affected the leaf δ^13^C of plants, and leaf δ^13^C increased with the increase in tree height.

Most studies show that climatic variables have important roles in shaping the patterns of leaf δ^13^C, in addition to the physiological characteristics of plants [12,13,14]. Zhou et al. [15] studied the relationship between δ^13^C and temperature and precipitation in temperate grassland plants in Inner Mongolia and found that the δ^13^C of all plants increased with increasing temperature, and decreased with increasing precipitation. Ma et al. [16] studied the spatial variation of stable carbon isotope composition in leaves of three species of Caragana in Northern China. The results showed that the δ^13^C in leaves of three species decreased significantly with the increase of MAP and RH, and increased with the increase of altitude and MAT. Liu et al. [17] studied the changes of δ^13^C in desert plant leaves in Northern China along climatic gradients and found that δ^13^C in plant leaves decreased with increasing precipitation. These studies are mainly about how environmental conditions affect plant leaf δ^13^C and how plants adapt to the dynamic changes of habitat by adjusting leaf δ^13^C. However, the effect of mycorrhizal type on plant leaf δ^13^C, especially the relationship between mycorrhizal type and plant leaf δ^13^C on a large scale, is still unknown.

The response of leaf δ^13^C traits to biological factors, such as mycorrhizal characteristics and their interrelationships, are of great significance to plants. Mycorrhizal fungi play a crucial role in the regulation of leaf δ^13^C [18,19]. These fungi are obligate symbionts that form a mutualistic relationship with plant roots, known as mycorrhiza. Previous studies have shown that mycorrhizal fungi affect the leaf δ^13^C of host plants by influencing the gas exchange parameters of plant leaves, such as photosynthetic rate, cellular CO_2_ concentration, and water use efficiency [20,21,22]. Although there are serval distinct types of mycorrhiza status in nature, we focused on arbuscular mycorrhizal (AM) and ectomycorrhizal (ECM) associations because they are the two most universal and best-studied types in terrestrial ecosystems. AM fungi are widespread, forming symbiotic associations with 85% of all terrestrial plants [23]. By contrast, ECM fungi are restricted to a smaller number of host plant species [24]. Vargas et al. [25] found that ecosystem CO_2_ fluxes were differently influenced by arbuscular and ectomycorrhizal-dominated vegetation types. Terrer et al. [26] suggested that mycorrhizal types can affect the plant nutrient acquisition strategies thus affecting the trade-off between plant and soil carbon storage under elevated CO_2_. The leaf δ^13^C and terrestrial ecosystem carbon cycle are closely linked.

However, the effects of different mycorrhizal fungi on the content of δ^13^C in the leaves of different life forms, and the relationship between δ^13^C and environmental factors, are still unclear.

Hence, we researched the difference in leaf δ^13^C between AM plants and ECM plants across life forms and further analyzed the relationship between environmental factors and leaf δ^13^C in AM plants and ECM plants across life forms. We considered that different types of mycorrhizal fungi have different effects on the physiological processes of plant leaves, especially on the photosynthetic processes of plants, and the geographical distribution patterns of different types of mycorrhizal fungi. We also propose two hypotheses: (1) the leaf δ^13^C value varies with mycorrhizal types; and (2) the leaf δ^13^C of different mycorrhizal types varies with environmental factors.

## 2. Materials and Methods

### 2.1. Assembly of Database

Most C_3_ plants are positive among land plants, and the δ^13^C of C_3_ plants can better help us understand the relationship between land plants and their environment [27,28]. In this study, leaf 1δ^13^C data of Chinese C_3_ terrestrial plants were obtained from the database constructed by Li et al. [7] along with the life form of each plant, photosynthesis type and environmental data such as mean annual precipitation (MAP, mm), relative humidity (RH, %), mean annual temperature (MAT, °C) and solar hours (SH, hours) for each sampling site. We established a new database of leaf δ^13^C of different mycorrhizal types of C_3_ plants in China based on the database constructed by Li et al. [7].

Based on the reference information provided by Li et al., we identified the specific plant species corresponding to each observation. The mycorrhizal type of each plant species was ascertained according to the published literature, especially by Wang and Qiu [29], Averill et al. [24], and Shi et al. [30]. We classified all the plants with typical AM and ECM structures as AM type and ECM type.

In order to compare the differences between AM and ECM in different plant life types, plants were subdivided into two subgroups based on their growth forms, i.e., woody species and herbaceous species. In this study, we refer to those woody plants with independent trunks as trees that occur from roots and have a distinct trunk and crown, usually higher than 6 m. Woody plants without a distinct trunk and in a clumped state are dwarf, usually less than 6 m, and are referred to as shrubs. Plants with herbaceous or fleshy stems with less developed woody parts, whose above-ground parts mostly die in the same year, are called herbaceous plants. The woody species were divided into two sub-sub groups, i.e., trees and shrubs; whereas the herbaceous species were divided into the annual herb and perennial herb. The herbaceous species included 53 annual herbs and 337 perennial herbs. Of these, 53 annual herbaceous plants and 336 perennial herbaceous plants belong to AM plants and only 1 perennial plant belongs to ECM plants. The trees were further divided into deciduous trees and evergreen trees; According to statistics, among the plants in the database, 817 species belong to the AM group, including 153 tree species, 275 shrub species, and 389 herb species; and 167 species belong to the ECM group, including 160 tree species, 6 shrub species, and 1 herb species. Since the number of shrub species and herb species in ECM plants was too little to compare, for the accuracy of the conclusion, we only compare the leaf δ13C of AM plants and ECM plants in four groups (i.e., total plants, trees, evergreen trees, deciduous trees).

### 2.2. Data Analysis

Carbon isotopic value is expressed as the standard notation relative to the Vienna Pee Dee Belemnite standard using the following equation: δ^13^C = (R_sample_/R_standard_ − 1) × 1000 (‰), where R_sample_ and R_standard_ are the ^13^C/^12^C ratios of the sample and the standard, respectively [31]. In our study, leaf δ^13^C values of C_3_ plants were obtained from the database established by Li et al. [7]. To compare the differences in leaf δ^13^C between AM plants and ECM plants, we calculated the means of leaf δ^13^C of AM plants and ECM plants for each group. The δ^13^C of AM plants and ECM plants variation in four groups was examined by one-way analysis of variance. Then, after having diagnosed the covariance of environmental factors, we attempted to establish the relationship between environmental factors and plant leaf δ^13^C using multiple regression analysis, comparing the adjusted R^2^ of the best multiple regression model to assess the relative importance of different environmental factors in determining plant leaf δ^13^C. All statistical analyses were conducted using the SPSS software (2012, ver. 22.0; SPSS Inc., Chicago, IL, USA).

We tested the relationship between environmental factors and leaf δ^13^C of two mycorrhizal-type plants individually in each group. The slopes of regressions with 95% confidence intervals were displayed with scatter plots.

## 3. Results

### 3.1. The Differences in Leaf δ^13^C between AM and ECM Plants

The results of the one-way analysis of variance revealed that the differences in leaf δ^13^C between AM and ECM plants varied with life forms. When all vegetation types were considered, the leaf δ^13^C did not vary between AM and ECM plants (Figure 1a).

For AM plants, the mean of leaf δ^13^C is −27.01‰ and the value in ECM plants is −27.12‰. But the leaf δ^13^C of AM and ECM plants varied significantly among trees (Figure 1b, *p* < 0.001). For AM plants, the mean of leaf δ^13^C is −28.23‰ and the value in ECM plants is −27.09‰. When only evergreen trees were considered, a one-way analysis of variance results revealed that there were significant variations of leaf δ^13^C between AM and ECM plants, leaf δ^13^C of ECM plants was significantly higher than that of AM plants (Figure 1c, *p* < 0.001). By contrast, there was no significant difference in δ^13^C between AM plants and ECM plants.

### 3.2. Variation of Leaf δ^13^C across Longitude and Latitude in AM and ECM Plants

For all plants, the leaf δ^13^C of AM and ECM both decreased significantly with the increase in longitude and there is a visible difference in leaf δ^13^C in AM and ECM plants (P_AM_ < 0.05; P_ECM_ < 0.001; Figure 2a). With the increase of longitude, the decreasing rate of δ^13^C value of ECM plant leaves was significantly higher than that of AM plant leaves (P_AM&ECM_ < 0.001). For all trees, the leaf δ^13^C of ECM plants decreased significantly with the increase in longitude (P_ECM_ < 0.001), and there were no significant interactions between the leaf δ^13^C of AM plants and the longitude (Figure 2b). For evergreen trees, the leaf δ13C in AM plants decreased significantly with the increase of longitude (P_ECM_ < 0.001; Figure 2c), yet no notable linear regression relation was found between leaf δ^13^C of AM plants and longitude. For deciduous trees, both the leaf δ^13^C in AM plants and ECM plants do not correlate with the longitude (Figure 2d).

When all vegetation types were considered, the leaf δ^13^C of AM and ECM both first increased and then decreased significantly with the increase of latitude (P_AM_ < 0.001; P_ECM_ < 0.001; Figure 3a). It is noteworthy that even though the two curves follow the same trend, the maximum value of leaf δ^13^C is different in latitude. In AM plants, the leaf δ^13^C had the highest value at the latitude of 42.10°. In ECM plants, the leaf δ^13^C had the highest value at the latitude of 37.56°. In total plants, the maximum value of δ^13^C in the leaves of AM plants was −21.05‰, which was higher than the maximum value of δ^13^C in the leaves of ECM plants (−23.88‰). When only trees were considered, the relationship between leaf δ^13^C of two kinds of mycorrhizal type plants and latitude in trees corresponded to that in total plants (P_AM_ < 0.001; P_ECM_ < 0.001; Figure 3b). In AM plants, the leaf δ^13^C had the highest value at the latitude of 45.08°. In ECM plants, the leaf δ^13^C had the highest value at the latitude of 36.71°. In trees, however, the difference between the leaf δ^13^C maxima of AM plants and ECM plants decreased significantly, and the leaf δ^13^C maxima of AM plants (−23.26‰) were higher slightly than those of ECM plants (−23.88‰). When only evergreen trees were considered, even though the two parabolas open in opposite directions, leaf δ^13^C of two kinds of mycorrhizal types have been rising as latitude within the scope of the study. In evergreen plants, the maximum value of leaf δ^13^C was −23.88‰ for ECM plants. In deciduous plants, the maximum value of leaf δ^13^C was −23.46‰ for AM plants.

### 3.3. Variation of Leaf δ^13^C with Environmental Factors in AM and ECM Plants

For all species, the leaf δ^13^C in AM and ECM plants both decreased significantly with the increase of MAP (P_AM_ < 0.001; P_ECM_ < 0.001), with a similar slope (Figure 4a). For all trees, the change of leaf δ^13^C in AM and ECM plants still had the same response to the change of MAP. However, the effect of MAP on leaf 13C of ECM plants was greater than that of AM plants. With the increase in rainfall, the decrease rate of leaf 13C of ECM plants was twice that of AM plants (Figure 4b). For evergreen trees, leaf δ^13^C showed similar relationships with MAP across mycorrhizal-type plants, with the slopes being −0.0018 (*p* < 0.001) in AM plants, and −0.0021 (*p* < 0.001) in ECM plants, respectively. But the effect of MAP on ECM plants (R^2^ = 0.27) explained 10% more of the variation in leaf δ^13^C than the effect of MAP on AM plants (R^2^ = 0.17, Figure 4c). For deciduous trees, the leaf δ^13^C is uncorrelated with MAP, whether in AM types of plants or ECM types of plants (Figure 4d). 

When all vegetation types were considered, leaf δ^13^C in AM plants and ECM plants both decreased significantly with increasing MAT (P_AM_ < 0.001; P_ECM_ < 0.05). The slope of leaf δ^13^C in AM plants across MAT is twice as large as it is in ECM plants (Figure 5a). When only evergreen trees were considered, leaf δ^13^C in AM plants and ECM plants was also positively related to MAT, while the response of leaf δ^13^C to MAT tended to be more sensitive in AM plants (slope = 0.10) in comparison to ECM plants (slope = 0.06; Figure 5c). By contrast, leaf δ^13^C in AM plants and ECM plants had no correlation with MAT, in deciduous trees (Figure 5d). 

In all vegetation type group, leaf δ^13^C tended to decrease with the increasing relative humidity, which is both significant in AM plants (*p* < 0.001) and in ECM plants (*p* < 0.001), but, the effect of relative humidity change on δ^13^C in ECM plant leaves was more obvious, and the value of R^2^ was higher (Figure 6a). In the trees group, leaf δ^13^C in AM plants and ECM plants both decreased significantly with increasing relative humidity (P_AM_ < 0.001, P_ECM_ < 0.001). The slope of leaf δ^13^C in AM plants across relative humidity is half as large as it is in ECM plants (Figure 6b). In the evergreen trees group, the leaf δ^13^C in AM plants and ECM plants both were negatively correlated with relative humidity (P_AM_ < 0.001, P_ECM_ < 0.001). Although the δ^13^C of AM plants is more sensitive to changes in RH than the δ^13^C of ECM plants, the effect of RH on ECM plants (R^2^ = 0.32) explained 5% more of the variation in leaf δ^13^C than the effect of MAP on AM plants (R^2^ = 0.27; Figure 6c). In the deciduous trees group, leaf δ^13^C in AM and ECM plants slightly increased with higher relative humidity, but the relationship was not significant (Figure 6d). 

For all vegetation types, for both AM plants and ECM plants, the leaf δ^13^C increased significantly with altitude (P_AM_ < 0.001; P_ECM_ < 0.05) and the change in altitude had the same effect on both species (Figure 7a). This correlation is also suited to that for trees, leaf δ^13^C in AM plants and ECM plants both related positively to altitude (P_AM_ < 0.001; P_ECM_ < 0.001). Besides, the effect of altitude on leaf δ^13^C of AM plants was similar to the effect of altitude on leaf δ^13^C of ECM plants (Figure 7b). For evergreen trees, leaf δ^13^C in AM plants and ECM plants both slightly increased with higher altitude but the relationship between δ^13^C in ECM plants and altitude was not significant (Figure 7c). By contrast, for deciduous trees, leaf δ^13^C tended to decrease with altitude and there was no correlation with altitude, whether the leaf δ^13^C in AM plants or ECM plants (Figure 7d).

When all vegetation types were considered, the leaf δ^13^C in AM plants and ECM plants both increased significantly as sunshine hours (P_AM_ < 0.001; P_ECM_ < 0.001) and were close in their sensitivity to sunshine hours responses (Figure 8a). When only trees were considered, leaf δ^13^C in AM plants and ECM plants were both positively correlated with sunshine hours (P_AM_ < 0.001; P_ECM_ < 0.001). The response of leaf δ^13^C to sunshine hours tended to be more sensitive in ECM plants (slope = 0.0014) in comparison to AM plants (slope = 0.0008). We further analyzed the linear relationship between leaf δ^13^C and sunshine hours of AM plants and ECM plants in evergreen trees and deciduous trees, respectively. The results showed that the effect of sunshine hours on leaf δ^13^C of AM plants and ECM plants in evergreen trees was significantly (P_AM_ < 0.05; P_ECM_ < 0.01) greater than the effect of sunshine hours on leaf δ^13^C of AM plants and ECM plants in deciduous trees (Figure 8c,d). In evergreen trees, the effect of sunshine hours on ECM plants (R^2^ = 0.1867) explained 10.6% more of the variation in leaf δ^13^C than AM plants, but with a similar slope (Figure 8c). 

### 3.4. Stepwise Regression Analysis of Leaf δ^13^C and Environmental Factors in AM and ECM Plants

The model summary was shown in Table 1. When all vegetation types were considered, in AM plants, the value of R^2^ was 0.266, which indicated that there were 26.6% changes in the response variable (leaf δ^13^C) because of changes in the combination of four controlled variables including Lat, Lon, MAT, and SH. Among four controlled variables that affected δ^13^C values, MAT was the most profound environment factor (β = −0.265, *p* < 0.001), while SH was the secondary environment factor (β = 0.245, *p* < 0.001). In ECM plants, the R^2^ value of the stepwise regression equation was higher than that of AM plants, which was 0.285. According to the results, the variation of leaf δ^13^C was mainly attributed to four variables, these variables were: RH (β = −0.449, *p* < 0.001), LAT (β = −0.413, *p* < 0.001), Altitude (β = 0.263, *p* < 0.001). When only trees were considered, the multivariate stepwise regression equation of leaf δ^13^C in AM plants only screened out LAT as a significant influencing factor (β = 0.448, *p* < 0.001). It explained 19.6 % of the total variation of leaf δ^13^C in AM plants. The multivariate stepwise regression equation of δ^13^C in leaves of ECM plants screened three significant influencing factors, namely Lat (β = −0.445, *p* < 0.001), RH (β = −0.459, *p* < 0.001), Altitude (β = 0.279, *p* < 0.001), which together explained 41.5 % of the total variation of δ^13^C in leaves. When only evergreen trees were considered, in AM groups, the best leaf δ^13^C model showed that Lat (β = 0.789, *p* < 0.001), Lon (β = −0.417, *p* < 0.001), and Altitude (β = −0.465, *p* < 0.001) in combination explained 43.2% of the total variation. Compared to this model, the leaf δ^13^C of ECM plants was explained by a set of three environmental factors including Lat (β = −1.773, *p* < 0.001), Lon (β = −1.483, *p* < 0.001), and RH (β = −0.583, *p* < 0.001), the explanation rate of the model is 49.4%.

## 4. Discussion

Here, we evaluated firstly the leaf δ^13^C and their relationship with environmental factors of C_3_ plants between AM and ECM types. The influence of mycorrhiza on plant development and its response to climate change vary with different mycorrhizal types, both at the individual level and at the ecological level [25,32]. Vargas et al. [25] suggested that ecosystem CO2 fluxes of ECM symbiosis tend to become dominant in woody vegetation types where interannual variation in ecosystem CO_2_ fluxes is primarily controlled by changes in temperature, whereas the AM symbiosis dominates grassland or woody vegetation types where interannual variation in CO_2_ fluxes was largely controlled by changes in precipitation. Shi et al. [30] presented that the world leaf economic spectrum traits were greatly linked with mycorrhizal traits and are woody plants, especially trees, have shorter leaf lifespans, lower leaf mass per area, and higher leaf nitrogen concentration, photosynthetic capacity, and dark respiration rate than nonwoody ones.

Our results further support this conclusion and point out that plant leaf δ^13^C is closely related to plant mycorrhizal types. The data presented here clearly show that variations in mean annual temperature (MAT), mean annual precipitation (MAP), and relative humidity (RH) are important environmental drivers for leaf δ^13^C, but influence AM and ECM-dominated vegetation types differently.

### 4.1. Overall Differences in Leaf δ^13^C between AM Plants and ECM Plants

Our analysis indicated that the arithmetic means of leaf δ^13^C in AM plants is −27.01‰ and ECM plants is −27.12‰, which were both nearly identical to the global average of leaf δ^13^C, −27.0‰. The latter was reported by Kohn [33], who collected leaf δ^13^C values from approximately 570 sites on a global scale. Moreover, our results were slightly higher than the global average (−27.25‰) reported by O’Leary [34].

Our results suggested that there were no differences in leaf δ^13^C between AM plants and ECM plants in total plants while the leaf δ^13^C of AM plants was less than ECM plants in trees (Figure 1a,b). Compared to the total plant group, we found leaf δ^13^C of AM plants decreased significantly but the leaf δ^13^C of ECM plants hardly change in the tree group. The analysis results show that this phenomenon was mainly caused by shrubs and herbs, because shrubs and herbs constitute the majority of AM plants, and the leaves of shrubs and herbs δ^13^C was greater than the leaves of trees δ^13^C. Previous studies presented that there were significant differences in leaf δ^13^C among different life forms [35].

After further analysis, we found that the differences in leaf δ^13^C between two mycorrhizal types of plants in evergreen trees were the main reason for the differences in trees (Figure 1c). The average value of leaf δ^13^C of AM plants was less than ECM plants in evergreen trees. Numerous studies have demonstrated that different mycorrhizal types differ in their use to improve nutrient uptake by plants; for example, AM mainly improves P nutrition of plants, while ECM facilitates N uptake by plants. Therefore, different mycorrhiza fungi also have different effects on the physiological processes of plants. Zhao et al. [36] found that the hydraulic conductivity of ECM trees was significantly higher than that of AM species, indicating that ECM species have higher photosynthetic rates. In summary, there are significant differences between mycorrhizal types in terms of their effects on plant physiological processes and nutrient cycling in the ecosystem. As an important characteristic of plant leaves, leaf δ^13^C is an important indicator to study the relationship between plants and their environment, and our results show that there are significant differences in leaf δ^13^C among different mycorrhizal types. This may be related to the different effects of different mycorrhizal types on photosynthetic rates as well as the stomatal conductance of their host plants. The exact reasons for this are to be further verified.

### 4.2. Differences in the Spatial Distribution of Leaf δ^13^C in AM Pants and ECM Plants

As an important characteristic of leaves, most studies demonstrated that leaf δ^13^C and geolocation information were related closely [37,38]. Li et al. [7] invested the spatial pattern of leaf δ^13^C values in China and found that leaf δ^13^C slightly decreased as longitude increased, but first increased, and then decreased as the latitude increased. Based on the study researched by Li et al. [7], we analyzed the relationship between leaf δ^13^C and longitude, and latitude, respectively, after classifying vegetation types as AM or ECM dominant. In our study, the leaf δ^13^C of AM plants and ECM plants both decreased significantly as longitude increased (except leaf δ^13^C of AM plants in trees) while the leaf δ^13^C of ECM plants was more sensitive to longitude change than AM plants (Figure 2a,b). The leaf δ^13^C of AM and ECM plants both first increased and then decreased as the latitude increased in total plants and tree groups. However, the maximum leaf δ^13^C of AM plants and ECM plants appeared at different latitudes. The latitude at which the maximum of leaf δ^13^C in AM plants occurs was greater than ECM plants in total plants and tree plants (Figure 3a,b). Our results also show that in different groups, latitude always has opposite effects on leaf δ^13^C of AM plants and ECM plants. It may be related to the global distribution of mycorrhizal fungi [39]. Due to the different selectivity of mycorrhizal fungi to host plants and adaptability to environmental conditions or the historical reasons in the evolutionary process, the distribution of mycorrhizal fungi in the natural ecosystem is different [40]. Numerous studies showed that mycorrhizal status significantly affects the responses of leaf characteristics to geographical distribution [28,40], our results further confirm this conclusion. Lu et al. [41] found that leaf ash concentration in arbuscular mycorrhizal plants was significant to be controlled by latitude and temperature factors, while ECM plants were not. Our results suggested that the spatial distribution of δ^13^C in plant leaves will be affected by plant mycorrhizal types consistent with previous studies. Due to the lack of relevant studies, the specific reasons need to be further explored.

### 4.3. Variations in Response of Leaf δ^13^C to Environmental Factors between AM Plants and ECM Plants

Water change has a lasting impact on spatial variation and the characteristics of vegetation, and maybe alter leaf δ^13^C of plants [8,42,43,44,45]. In our study, we analyzed and quantified the relationship between water availability (including MAP and RH) and leaf δ^13^C in AM plants and ECM plants based on our database (Figure 4 and Figure 6). Our results showed that leaf δ^13^C in AM and ECM plants both decreased as water availability increased (including MAP and RH), which indicated that the higher precipitation, the lower δ^13^C. Meanwhile, the results of step-by-step analysis showed that the relative humidity only affected the leaf δ^13^C of ECM plants, which further explained that the leaf δ^13^C of different mycorrhizal plants had different responses to water conditions (Table 1). The leaf δ^13^C of ECM plants has a higher change and explain rate than AM plants under changing water availability, both in total plants and trees (Figure 6a,b). This result indicated that mycorrhizal fungi can affect the response of leaf δ^13^C to changes in water availability. This may be due to the different sensitivity of the two kinds of mycorrhizal fungi to water changes [5,46]. Most studies suggested that ECM plants were more sensitive to water availability than AM plants [37,47]. Shi et al. [47] compared separately the net primary productivity of AM and ECM type forests and found that ECM type forests were more sensitive to precipitation changes than AM type forests. Zhao et al. [36] found that ECM trees have stronger drought resistance ability and higher water use efficiency compared with AM trees under the background of increasing drought in subtropical forest. Studies have shown that mycorrhiza can affect the stomatal conductance of plants [48,49]. The response of leaf characteristics to changes in water availability was affected by mycorrhizal types [30,41]. Lu et al. [41] studied the responses of leaf ash concentration of AM plants and ECM plants to climate change and found that the response of AM plants was more susceptible to mean annual precipitation with a 1.61 times response amplitude compared to ECM plants. All the above studies showed that leaf characteristics of different mycorrhizal types of plants were different in response to climate change. As one of the characteristics of plant leaves, the leaf δ^13^C of different mycorrhizal plants has different responses to the change in water availability.

In our study, we found a negative liner for the relationship of leaf δ^13^C in AM and ECM plants with MAT, but in the tree leaf δ^13^C of AM, plants were more sensitive to the changes in MAT than ECM plants trees (Figure 5a,b). Temperature affects the δ^13^C of plants mainly by affecting enzymes involved in photosynthesis [50]; AM and ECM both can promote photosynthesis in plants [51,52]. Wang et al. [53] found that AM inoculation had a significant effect on the photosynthetic of Sinocalycanthus Chinensis under simulated warming conditions. Compared to ECM, AM was more closely related to the photosynthesis of plants. This may be related to the different metrological characteristics of plant leaves of different mycorrhizal types. Studies have shown that AM plays the most significant role in improving the P nutrient status of plants, while ECM plays a greater role in the N nutrient absorption of plants [24]. As an important component of Rubisco and energy substance ATP, phosphorus plays an important role in plant photosynthesis. As this study is the first time to explore, the specific reasons need to be studied in the future.

Although the δ^13^C in AM and ECM plants have different responses to changes in temperature and water availability, they have the same response to changes in other environmental factors (e.g., altitude, sunshine hours). In our study, we found that the leaf δ^13^C of AM plants and ECM plants both increased weakly as altitude and sunshine hours increased (Figure 7 and Figure 8). This result indicated that certain environmental factors affect mycorrhizal fungi in the same way.

Overall, although plant life forms and environmental conditions are inherently variable, the variation of plant leaf δ^13^C along environmental gradients offers one way to evaluate potential plant responses to climate change. The different responses of the leaf δ^13^C of mycorrhizal types to climate change might also provide a reference for future studies, simulating the response of vegetation distribution to climate change. As this study is the first to explore this, the specific reasons need to be studied in the future.

## 5. Conclusions

All life forms will affect the leaf δ^13^C content of plants with different mycorrhizal types. The effect of arbuscular and ectomycorrhizal on the responses of leaf δ^13^C to changes in environmental factors was different. Further analysis showed that the results showed that the response of AM plant leaf δ^13^C and ECM plant leaf δ^13^C to climate variations depended on plant life forms. This study initially explored the effect of mycorrhizal on leaf δ^13^C. It also provides data in support of future exploration of the response of leaf δ^13^C in AM and ECM plants to changes in climate and environmental factors. Findings showed that the responses of leaf δ^13^C to changes in environmental factors are differentially affected by arbuscular and ectomycorrhizal types; the leaf δ^13^C of AM plants was mainly affected by temperature; while the leaf δ^13^C of ECM plants was more sensitive to moisture content. Our results have important implications for understanding the relationship between leaf δ^13^C and climatic factors, and the climatic and environmental significance indicated by plant leaf δ^13^C. This is a previously unrecognized study that has important implications for our understanding of the impacts of arbuscular and ectomycorrhizal on plants’ adaptation to climate change.

## Figures and Tables

**Figure 1 plants-11-03236-f001:**
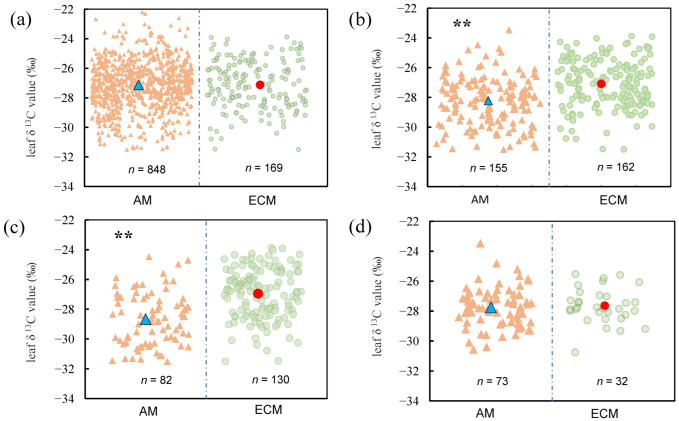
Leaf δ^13^C values of AM and ECM plants with different life forms. (**a**) Total plants, include all plants in the database without distinguishing their life forms (**b**) Trees, only contain all the tree species in the database (**c**) Evergreen trees, only contain all evergreen trees of the tree species (**d**) Deciduous trees, only contain all deciduous trees of the tree species. All yellow triangles represent AM plants and all green rectangles represent ECM plants. The blue triangle represents the average of leaf δ^13^C values of AM plants and the red circle represents the average of leaf δ^13^C in ECM plants. The asterisks above the line bars show the results of one-way ANOVA at the level of *p* < 0.05. Two asterisks indicate extremely significant differences.

**Figure 2 plants-11-03236-f002:**
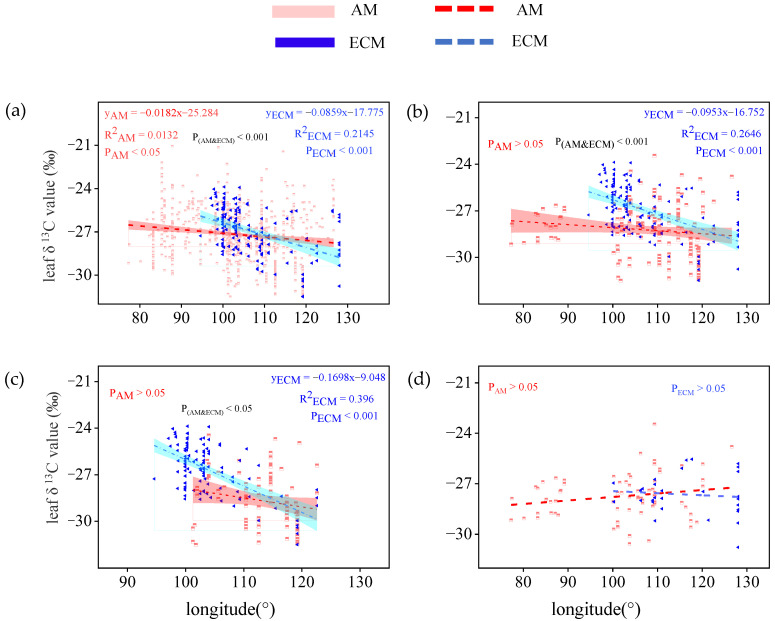
Variation of leaf δ^13^C in AM plants and ECM plants with different life forms along longitude. (**a**) Total plants, include all plants in the database without distinguishing their life forms (**b**) trees, only contain all the tree species in the database (**c**) evergreen trees, only contain all evergreen trees of the tree species (**d**) deciduous trees, only contain all deciduous trees of the tree species. The red broken line means AM, the blue broken line means ECM. The red and cyan bands represent, respectively, the prediction intervals of AM and ECM plants. P(AM and ECM) represents the significant difference in the slope of the two regression lines.

**Figure 3 plants-11-03236-f003:**
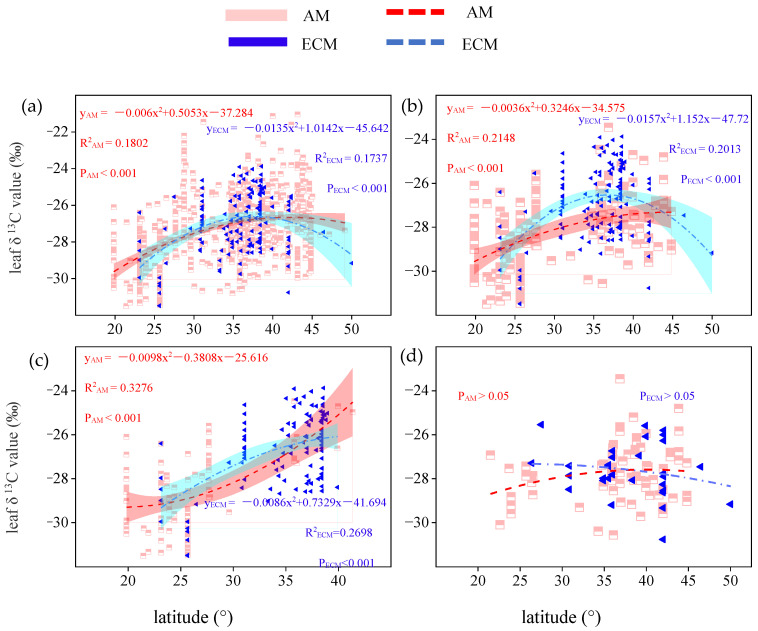
Variation of leaf δ^13^C in AM plants and ECM plants with different life forms along latitude. (**a**) Total plants, include all plants in the database without distinguishing their life forms (**b**) trees, only contain all the tree species in the database (**c**) evergreen trees, only contain all evergreen trees of the tree species. (**d**) deciduous trees, only contain all deciduous trees of the tree species. The red broken line means AM, the blue broken line means ECM. The red and cyan bands represent, respectively, the prediction intervals of AM and ECM plants.

**Figure 4 plants-11-03236-f004:**
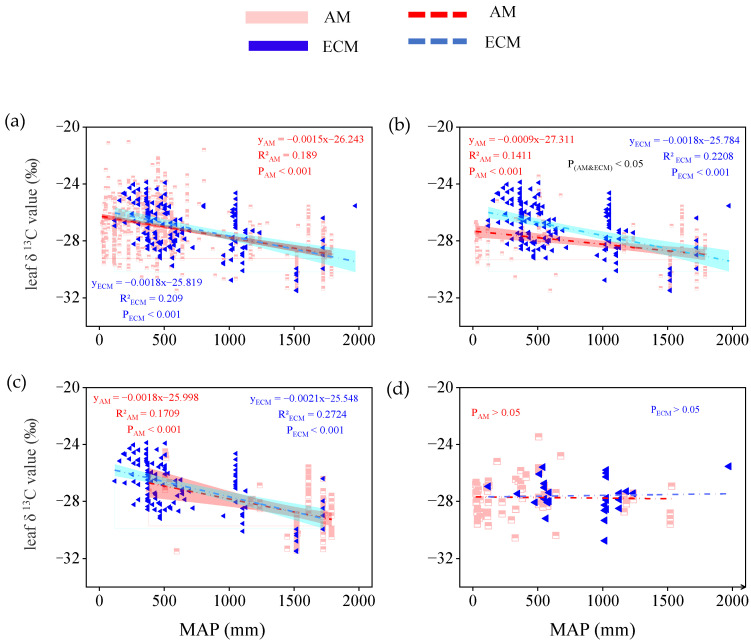
The relationship between leaf δ13C in AM plants and ECM plants with different life forms and mean annual precipitation (MAP, mm). (a) Total plants, include all plants in the database without distinguishing their life forms. (b) Trees, only contain all the tree species in the database.(c) Evergreen trees, only contain all evergreen trees of the tree species. (d) Deciduous trees, only contain all deciduous trees of the tree species. The red broken line means AM, the blue broken line means ECM. The red and cyan bands represent, respectively, the prediction intervals of AM and ECM plants. P(AM and ECM) represents the significant difference in the slope of the two regression lines.

**Figure 5 plants-11-03236-f005:**
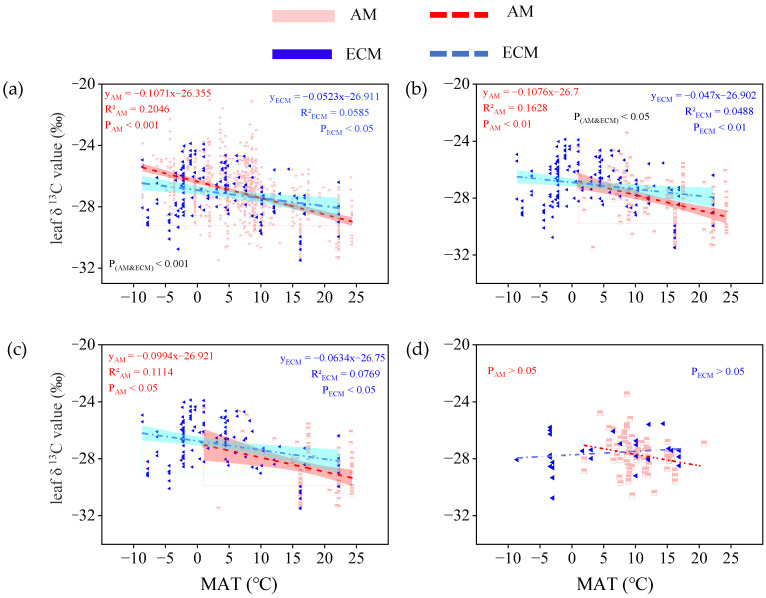
The relationship between leaf δ^13^C in AM plants and ECM plants with different life forms and mean annual temperature (MAT, °C). (**a**) Total plants, include all plants in the database without distinguishing their life forms. (**b**) Trees, only contain all the tree species in the database. (**c**) Evergreen trees, only contain all evergreen trees of the tree species. (**d**) Deciduous trees, only contain all deciduous trees of the tree species. The red broken line means AM, the blue broken line means ECM. The red and cyan bands represent, respectively, the prediction intervals of AM and ECM plants. P(_AM and ECM_) represents the significant difference in the slope of the two regression lines.

**Figure 6 plants-11-03236-f006:**
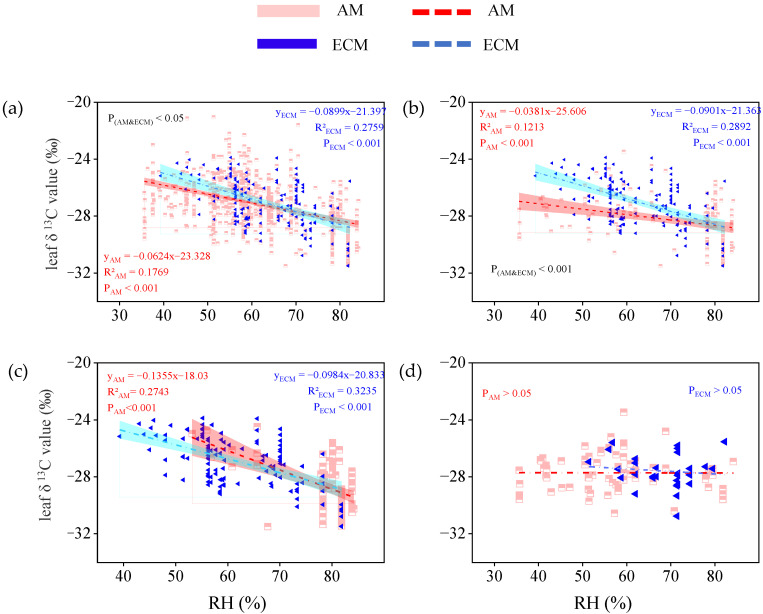
The relationship between leaf δ^13^C in AM plants and ECM plants with different life forms and relative humidity (RH, %). (**a**) Total plants, include all plants in the database without distinguishing their life forms. (**b**) Trees, only contain all the tree species in the database. (**c**) Evergreen trees, only contain all evergreen trees of the tree species. (**d**) Deciduous trees, only contain all deciduous trees of the tree species. The red broken line means AM, the blue broken line means ECM. The red and cyan bands represent, respectively, the prediction intervals of AM and ECM plants. P(AM and ECM) represents the significant difference in the slope of the two regression lines.

**Figure 7 plants-11-03236-f007:**
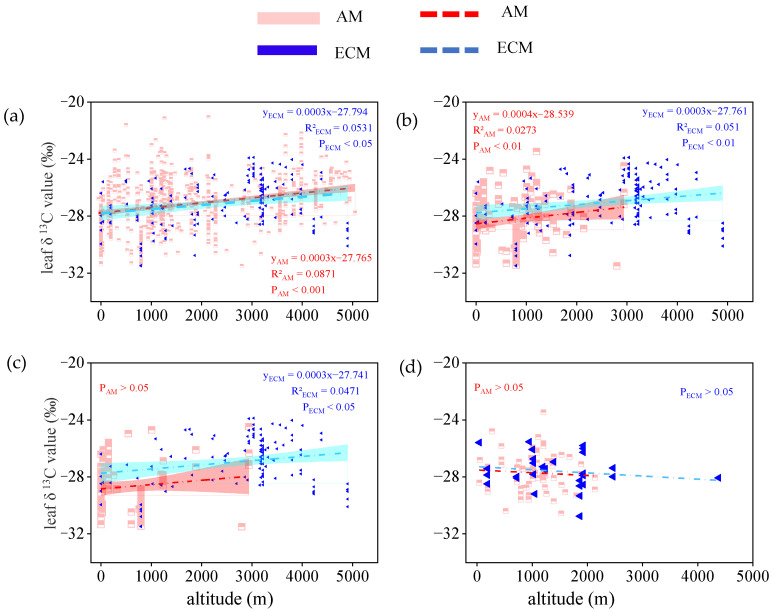
The relationship between leaf δ^13^C in AM plants and ECM plants with different life forms and altitudes (m). (**a**) Total plants, include all plants in the database without distinguishing their life forms. (**b**) Trees, only contain all the tree species in the database. (**c**) Evergreen trees, only contain all evergreen trees of the tree species. (**d**) Deciduous trees, only contain all deciduous trees of the tree species. The red broken line means AM, the blue broken line means ECM. The red and cyan bands represent, respectively, the prediction intervals of AM and ECM plants. P(_AM and ECM_) represents the significant difference in the slope of the two regression lines.

**Figure 8 plants-11-03236-f008:**
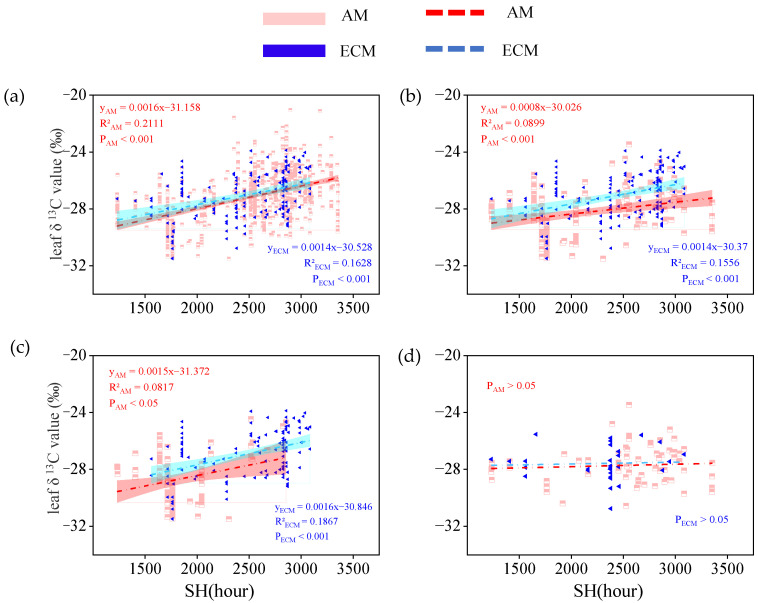
The relationship between leaf δ^13^C in AM plants and ECM plants with different life forms and sunshine hours (SH, hours). (**a**) Total plants, include all plants in the database without distinguishing their life forms. (**b**) Trees, only contain all the tree species in the database. (**c**) Evergreen trees, only contain all evergreen trees of the tree species. (**d**) Deciduous trees, only contain all deciduous trees of the tree species. The red broken line means AM, the blue broken line means ECM. The red and cyan bands represent, respectively, the prediction intervals of AM and ECM plants. P(_AM and ECM_) represents the significant difference in the slope of the two regression lines.

**Table 1 plants-11-03236-t001:** Results of the multiple regression analyses for predicting the leaf δ^13^C of AM and ECM plants with the combination of all traits.

Groups	Mycorrhizal Types	Standardized Coefficients							R^2^	Sig
		LAT	LON	MAP	MAT	RH	Altitude	SH		
Total plants	AM	0.182	0.120	-	−0.265	-	-	0.245	0.266	***
	ECM	−0.413	-	-	-	−0.449	0.263	-	0.285	***
Trees	AM	0.448	-	-	-	-	-	-	0.196	***
	ECM	−0.445	-	-	-	−0.459	0.279	-	0.415	***
Evergreen trees	AM	0.789	−0.417	-	-	-	−0.465	-	0.432	***
	ECM	−1.773	−1.483	-	-	−0.583	-	-	0.494	***

AM: arbuscular mycorrhiza, ECM: ectomycorrhiza, LAT: latitude, LON: longitude, MAP: mean annual precipitation, MAT: mean annual temperature, RH: relative humidity, SH: sun hours. - means that the environmental factors were not included in the best model. *** *p* < 0.001.

## Data Availability

The raw data for this study available via https://figshare.com/s/fcfdd37541c8ba1b8278 (accessed on 3 April 2022).

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
