# Peer review of "The Variations of Leaf δ^13^C and Its Response to Environmental Changes of Arbuscular and Ectomycorrhizal Plants Depend on Life Forms"

_plants, 2022, doi:10.3390/plants11233236_

Round 1
Reviewer 1 Report
This study investigated the variation in leaf δ13C between AM plants and ECM plants. They found that AM and ECM trees have different leaf δ13C value and the significant differences were also found between AM and ECM evergreen trees. The differences in leaf δ13C between AM and ECM plants related closely to life forms. The authors also analysis the correlation between leaf δ13C values and several environmental factors.
I see several flaws in the data analysis. Linear regression is one of the most common techniques of regression analysis. However, for complex connections between data, the relationship might be explained by more than one variable. In this case, an analyst should use multiple regression which attempts to explain a dependent variable using more than one independent variable. I recommend to reanalyze the data by multiple regression.
Author Response
Dear Editor:
On behalf of my co-authors, we thank you very much for giving us an opportunity to revise our manuscript, we appreciate you and reviewers very much for your positive and constructive comments and suggestions on our manuscript entitled “The variations of leaf δ13C and its response to environmental changes of arbuscular and ectomycorrhizal plants depend on life forms”. (ID:plants-1967535).
We have studied reviewers’ comments carefully and have made revision in the manuscript. And We have also answered the reviewers’ comments by point to point. Please find the following “Response to Reviewers”. We have tried our best to revise our manuscript according to the comments. We would like to express our great appreciation to you and reviewers for comments on our paper. Looking forward to hearing from you.
Thank you and best regards.
Yours sincerely,
Zhaoyong SHI
Responses to Reviewer
Yes Can be improved Must be improved Not applicable
Does the introduction provide sufficient background and include all relevant references?
( ) ( ) (x) ( )
Response: Accepted. The introduction had been revised to make it more complete. We would like to revise further it if you believe it is necessary. The detail changes are following:
To Add some description:
Lines 33-36: Leaf carbon isotope discrimination (δ13C) plays an important role in our understanding of the relationship between plants and the environment, It not only provides a series of climate and environmental information related to plant growth processes but also can indicate how plants interact with the environment and respond[1-4]. Thus, the relationship between leaf δ13C and the environment has aroused interest [5-8].
Lines 39-40: The study found significant differences in δ13C among different species of R. natans, which may be caused by stomatal limitation[4].
Lines 43-44: He et al. [11] found that tree height affected the leaf δ13C of plants, and leaf δ13C increased with the increase in tree height.
Lines 45-55: Most studies show that climatic variables have important roles in shaping patterns of leaf δ13C in addition to the physiological characteristics of plants [12-14]. Zhou et al. [15] studied the relationship between δ13C and temperature and precipitation in temperate grassland plants in Inner Mongolia and found that δ13C of all plants increased with increasing temperature and decreased with increasing precipitation. Ma et al. [16] studied the spatial variation of stable carbon isotope composition in leaves of three species of Caragana in northern China. The results showed that δ13C in leaves of three species decreased significantly with the increase of MAP and RH, and increased with the increase of altitude and MAT. Liu et al. [17] studied the changes of δ13C in desert plant leaves in northern China along climatic gradients and found that δ13C in plant leaves decreased with increasing precipitation.
Lines 60-61: The response of leaf δ13C traits to biological factors such as mycorrhizal characteristics and their interrelationships are of great significance to plants.
Lines 76-78: in leaves of different life forms and the relationship between δ13C and environmental factors are still unclear.
Lines 81-84: Considering that different types of mycorrhizal fungi have different effects on the physiological processes of plant leaves, especially on the photosynthetic processes of plants, and the geographical distribution patterns of different types of mycorrhizal fungi.
To make the background more complete, we added some references are as follows:
- Wang, N.; Xu, S.S.; Jia, X.; Gao, J.; Zhang, W.P.; Qiu, Y.P.; Wang, G.X. Variations in foliar stable carbon isotopes among functional groups and along environmental gradients in China- A meta- analysis. Plant Biology. 2013, 15, 144– 151.
- Zhou, Y.C.; Li, H.B.; Xu, X.Y.; Li Y.H. Responses of carbon isotope composition of common C3and C4plants to climatic factors in temperate grasslands. Sustainability. 2022, 14, 7311.
- Mueller, K.E., Blumenthal, D.M., Pendall, E., Carrillo, Y., Dijkstra, F.A., Williams, D.G., Follett R.F., Morgan, J.A. Impacts of warming and elevated CO2on a semiarid grassland are non‐additive, shift with precipitation, and reverse over time. Ecology Letters. 2016, 19, 956-966.
- Li, Z.Q.; Yang, L.; Lu, W.; Guo, X.S.; Xu, J.; Yu, D. Spatial patterns of leaf carbon, nitrogen stoichiometry and stable carbon isotope composition of Ranunculus natans C.A. Mey. (Ranunculaceae) in the arid zone of northwest China. Ecol. Eng. 2015, 77, 9-17.
- Chen, Z.X.; Wang, G.A.; Jia, Y.F. Foliar δ13C showed no altitudinal trend in an arid region and atmospheric pressure exerted a negative effect on plant δ13C. Front Plant Sci. 2017, 8, 1070.
- Yoneyama, T.; Okada, H.; Ando, S. Seasonal variations in natural δ13C abundances in C3 and C4plants collected in Thailand and the Philippines. Soil Sci. Plant Nutr. 2010, 56, 422-426.
- 9. Stein, R.A.; Sheldon, N.D.; Smith, S.Y. C3plant carbon isotope discriminatiodoes not respond to CO2concentration on decadal to centennial timescales. New `Phytol. 2020, 229, 2576-2585.
- 1 He, C.X.; Li, J.Y.; Zhou, P.; Guo, M.; Zheng, Q.S. Changes of leaf morphological, anatomical structure and carbon isotope ratio with the height of the Wangtian tree (Parashorea chinensis) in Xishuangbanna, China. J Integr Plant Biol. 2008, 50, 168-173.
- Ale, R.; Zhang, L.; Li, X.; Raskoti, B.B.; Pugnaire, F.I.; Luo, T.X. Leaf delta δ13C as an indicator of water availability along elevation gradients in the dry Himalayas. Ecol. Indic. 2018, 94, 266-273.
- 1 Zhou, Y.C.; Zhang, W.B.; Cheng, X.L.; Harris, W.; Schaeffer, S.M; Xu, X.Y.; Zhao, B. Factors affecting δ13C enrichment of vegetation and soil in temperate grasslands in Inner Mongolia, China. J. Soils Sediments. 2019, 19, 2190-2199.
- 1 Ma, F.; Liang, W.Y.; Zhou, Z.N.; Xiao, G.J.; Liu, J.L.; He, J.; Jiao, B.Z.; Xu, T.T. Spatial variation in leaf stable carbon isotope composition of three caragana species in northern China. Forests. 2018, 9, 297.
- 1 Liu, J.; Su, Y.G.; Li, Y.; Huang, G. Shrub colonization regulates δ13C enrichment between soil and vegetation in deserts by affecting edaphic variables. Catena. 2021, 203, 105365.
Are all the cited references relevant to the research?
(x) ( ) ( ) ( )
Response: Thank the expert for your approval.
Is the research design appropriate?
( ) (x) ( ) ( )
Response: Accepted. Thank for your reminding. We add a new analysis method (multiple regression analysis) to the Materials Method section and add a detailed description of the method.
Are the methods adequately described?
( ) ( ) (x) ( )
Response: Accepted. We add a new analysis method (multiple regression analysis) to the Materials Method section and add a detailed description of the method.
Our revised content is as follows:
Lines 141-145: Carbon isotopic value is expressed as the standard notation relative to the Vienna Pee Dee Belemnite standard using the following equation: δ13C=(Rsample/Rstandard-1) × 1,000 (‰), where Rsample and Rstandard are the 13C/12C ratios of the sample and the standard, respectively[32]. In our study, leaf δ13C values of C3 plants were obtained from the database established by Li et al. [7]
Lines 148-152: Then, after having diagnosed the covariance of environmental factors, we attempted to establish the relationship between environmental factors and plant leaf δ13C using multiple regression analysis, comparing the adjusted R2 of the best multiple regression model to assess the relative importance of different environmental factors in deter-mining plant leaf δ13C.
Are the results clearly presented?
( ) ( ) (x) ( )
Response: Accepted. Thank for your Constructive comments. We have carefully revised the results section and added table1 to make the results more accurate. To better present our results, we also added Table1. It’s following:
The analysis of multiple regression analysis was conducted by SPSS software (version 22.0).
Table 1. results of the multiple regression analyses for predicting the leaf δ13C of AM and ECM plants with the combination of all traits
|
Groups |
Mycorrhizal types |
Standardized Coefficients |
R2 |
Sig |
||||||
|
LAT |
LON |
MAP |
MAT |
RH |
Altitude |
SH |
||||
|
Total plants |
AM |
0.182 |
0.120 |
- |
-0.265 |
- |
- |
0.245 |
0.266 |
*** |
|
ECM |
-0.413 |
- |
- |
- |
-0.449 |
0.263 |
- |
0.285 |
*** |
|
|
Trees |
AM |
0.448 |
- |
- |
- |
- |
- |
- |
0.196 |
*** |
|
ECM |
-0.445 |
- |
- |
- |
-0.459 |
0.279 |
- |
0.415 |
*** |
|
|
Evergreen trees |
AM |
0.789 |
-0.417 |
- |
- |
- |
-0.465 |
- |
0.432 |
*** |
|
ECM |
-1.773 |
-1.483 |
- |
- |
-0.583 |
- |
- |
0.494 |
*** |
|
- means that the environmental factors were not included in the best model. *** P<0.001
Add the following:
Lines 180-182: leaf δ13C of ECM plants was significantly higher than that of AM plants (Figure 1c, P<0.001). By contrast, , there was no significant difference in δ13C between AM plants and ECM plants.
Lines 199-201: With the increase of longitude, the decreasing rate of δ13C value of ECM plant leaves was significantly higher than that of AM plant leaves (PAM&ECM<0.001) .
Lines 231-233: In trees, however, the difference between the leaf δ13C maxima of AM plants and ECM plants decreased significantly, and the leaf δ13C maxima of AM plants (-23.26‰) were higher slightly than those of ECM plants (-23.88‰)
Lines 236-237: In evergreen plants, the maximum value of leaf δ13C was -23.88‰ for ECM plants. In deciduous plants, the maximum value of leaf δ13C was -23.46‰ for AM plants.
Lines 247-249: However, the effect of MAP on leaf δ13C of ECM plants was greater than that of AM plants. With the increase in rainfall, the decrease rate of leaf δ13C of ECM plants was twice that of AM plants (Figure 4b).
Lines 282-283: but, the effect of relative humidity change on δ13C in ECM plant leaves was more obvious, and the value of R2 was higher (Figure 6a).
Lines 337-356:
3.4. Stepwise regression analysis of Leaf δ13C and environmental factors in AM and ECM Plants
The model summary was shown in Table 1. When all vegetation types were considered, in AM plants, the value of R2 was 0.266, which indicated that there were 26.6% changes in the response variable (leaf δ13C) because of changes in the combination of four controlled variables including Lat, Lon, MAT, and SH. Among four controlled variables that affected δ13C values, MAT was the most profound environment factor (β=-0.265, P<0.001), while SH was the secondary environment factor (β=0.245, P< 0.001). In ECM plants, the R2 value of the stepwise regression equation was higher than that of AM plants, which was 0.285. According to the results, the variation of leaf δ13C was mainly attributed to four variables, these variables were: RH (β=-0.449, P<0.001), LAT(β=-0.413, P<0.001), Altitude(β=0.263, P<0.001). When only trees were considered, the mul-tivariate stepwise regression equation of leaf δ13C in AM plants only screened out LAT as a sig-nificant influencing factor (β=0.448, P<0.001), It explained 19.6 % of the total variation of leaf δ13C in AM plants. The multivariate stepwise regression equation of δ13C in leaves of ECM plants screened three significant influencing factors, namely Lat (β=-0.445, P<0.001 ), RH (β =-0.459, P<0.001), Altitude (β=0.279, P<0.001), which together explained 41.5 % of the total variation of δ13C in leaves. When only evergreen trees were considered, in AM groups, the best leaf δ13C model showed that Lat(β=0.789, P< 0.001), Lon(β=-0.417, P<0.001), and Altitude(β=-0.465, P<0.001) in combination explained 43.2% of the total variation. Compared to this model, the leaf δ13C of ECM plants was explained by a set of three environmental factors including Lat(β=-1.773, P<0.001), Lon(β=-1.483, P<0.001), and RH(β=-0.583, P<0.001), the explanation rate of the model is 49.4%
Table 1. results of the multiple regression analyses for predicting the leaf δ13C of AM and ECM plants with the combination of all traits
|
Groups |
Mycorrhizal types |
Standardized Coefficients |
R2 |
Sig |
||||||
|
LAT |
LON |
MAP |
MAT |
RH |
Altitude |
SH |
||||
|
Total plants |
AM |
0.182 |
0.120 |
- |
-0.265 |
- |
- |
0.245 |
0.266 |
*** |
|
ECM |
-0.413 |
- |
- |
- |
-0.449 |
0.263 |
- |
0.285 |
*** |
|
|
Trees |
AM |
0.448 |
- |
- |
- |
- |
- |
- |
0.196 |
*** |
|
ECM |
-0.445 |
- |
- |
- |
-0.459 |
0.279 |
- |
0.415 |
*** |
|
|
Evergreen trees |
AM |
0.789 |
-0.417 |
- |
- |
- |
-0.465 |
- |
0.432 |
*** |
|
ECM |
-1.773 |
-1.483 |
- |
- |
-0.583 |
- |
- |
0.494 |
*** |
|
- means that the environmental factors were not included in the best model. *** P<0.001
Are the conclusions supported by the results?
( ) ( ) (x) ( )
Response: Accepted. We have modified the conclusion section to make the conclusion more complete.
The main content of the conclusion:
- The variations of leaf δ13C in AM and ECM plants are different under differernt life forms
- The effects of climateon leaf δ13C are closely related to mycorrhizal types
- The leaf δ13C of AM plants was mainly affected by temperature, while the leaf δ13C of ECM plants was more sensitive to moisture content.
If there are any problems, we will make further modification.
Comments and Suggestions for Authors:
This study investigated the variation in leaf δ13C between AM plants and ECM plants. They found that AM and ECM trees have different leaf δ13C value and the significant differences were also found between AM and ECM evergreen trees. The differences in leaf δ13C between AM and ECM plants related closely to life forms. The authors also analysis the correlation between leaf δ13C values and several environmental factors.
I see several flaws in the data analysis. Linear regression is one of the most common techniques of regression analysis. However, for complex connections between data, the relationship might be explained by more than one variable. In this case, an analyst should use multiple regression which attempts to explain a dependent variable using more than one independent variable. I recommend to reanalyze the data by multiple regression.
Thanks to the reviewer for your efforts and comments in our manuscript. Thank the expert for your interest in our study. Stable carbon isotope composition (δ13C)
provides an integrated measurement of internal plant physiological and external environmental properties helping us understand the response of plants to environmental change. Different mycorrhizal plants have different responses to environmental changes. Hence, we investigated the effect of different mycorrhizal types on leaf δ13C under climate change. This study provides a comprehensive approach to studying the characteristics of leaf δ13C in different mycorrhizal plants under different life forms. We hope to provide valuable information for future studies on the effects of different mycorrhiza on leaf δ13C. We have read a lot of articles on the influencing factors of leaf δ13C, and found that these studies mainly use linear regression as the main analysis method(Zhang et al., 2015; Mueller et al., 2016; Li et al., 2017). Therefore, we also use linear regression to analyze the relationship between leaf δ13C and the environment in different mycorrhizal plants. Following your advice, we used multiple regression to analyze the data and correct the results to make them more concise and accurate. The effects of multiple environmental factors on δ13C in plant leaves were analyzed (See the table.1). Please refer to the modified version for detailed modification. Thank the reviewer for her/his efforts and comments in our manuscript and for giving us the opportunity to revise it.
Thank you for your comments on our manuscript again. We have tried our best to modify the manuscript. We will make further modifications to improve the quality of the manuscript if there are any problems.

Reviewer 2 Report
The following issues can be addressed in the present paper to improve its quality:
1. Line 88: Please explain why it is important to concentrate on the C3 plant for the present
study,
2. To provide further clarification, details and active database links can be provided.
3. Line 93-94: It is necessary to provide a thorough explanation of the procedure utilised to
identify the particular plant species.
4. Line 107-108: Data of annual herb and perennial herb is missing. Please elaborate.
5. Lines 114–115: Please describe the steps taken to estimate the leaf δ13C.
6. Analysis of leaf δ13C in AM plants and ECM plants for herbs is missing (Figure 1 to
Figure 8).
7. Define the significance of R 2 with respect to the present study and maintain consistency
when presenting it, as in Figure 2, where it is referred to as the R 2, and in lines 183, 184
shown as the R 2.
8. Line 160: Kindly give a comparative study and comment on the highest value of leaf
δ13C in each living form i.e. (a) Total plants, (b) Trees, (c) Evergreen trees, (d)
Deciduous trees.
Author Response
Dear Editor:
On behalf of my co-authors, we thank you very much for giving us an opportunity to revise our manuscript, we appreciate you and reviewers very much for your positive and constructive comments and suggestions on our manuscript entitled “The variations of leaf δ13C and its response to environmental changes of arbuscular and ectomycorrhizal plants depend on life forms”. (ID:plants-1967535).
We have studied reviewers’ comments carefully and have made revision in the manuscript. And We have also answered the reviewers’ comments by point to point. Please find the following “Response to Reviewers”. We have tried our best to revise our manuscript according to the comments. We would like to express our great appreciation to you and reviewers for comments on our paper. Looking forward to hearing from you.
Thank you and best regards.
Yours sincerely,
Zhaoyong SHI
Response to Reviewer
English language and style
( ) English very difficult to understand/incomprehensible
( ) Extensive editing of English language and style required
( ) Moderate English changes required
(x) English language and style are fine/minor spell check required
( ) I don't feel qualified to judge about the English language and style
Response: Thank for the expert approval .We have polished the language from beginning to end. If there are any problems, we will make further modification.
Yes Can be improved Must be improved Not applicable
Does the introduction provide sufficient background and include all relevant references?
( ) (x) ( ) ( )
Response: Accepted. Thank for your reminding. We have carefully reviewed the introduction section and have carefully made modifications in the expectation of providing sufficient background information and all relevant references.
Are all the cited references relevant to the research?
( ) (x) ( ) ( )
Response: Accepted. Thank for your reminding. We have carefully reviewed the reference section. We do our best to adjust and add relevant references. if there are still inadequacies, we would like to revise it further for improvement the quality of our manuscript if you consider it as necessary.
Is the research design appropriate?
( ) ( ) (x) ( )
Response: Accepted. We have adjusted and supplemented the research design part. In addition, adding relevant data to make the experimental content more complete.
Our revised content is as follows:
Most C3 plants are positive among land plants, and the δ13C of C3 plants can better help us understand the relationship between land plants and their environment[29 30]. In this study, leaf δ13C data of Chinese C3 terrestrial plants were obtained from the database constructed by Li et al [7] along with the life form of each plant, photosynthesis type and environmental data such as mean annual precipitation (MAP, mm), relative humidity (RH, %), mean annual temperature (MAT, °C) and solar hours (SH, hours) for each sampling site. We established a new database of leaf δ13C of different mycorrhizal types of C3 plants in China based on the database constructed by Li et al [7].
Based on the reference information provided by Li et al. we identified the specific plant species corresponding to each observation. The mycorrhizal type of each plant species was ascertained according to the published literature, especially by Wang and Qiu [31], Averill et al. [24], and Shi et al. [28]. We classified all the plants with typical AM and ECM structures as AM type and ECM type.
In order to compare the differences between AM and ECM in different plant life types, plants were subdivided into two subgroups based on their growth forms, i.e., woody species and herbaceous species. In this study, we refer to those woody plants with independent trunks as trees that occur from roots and have a distinct trunk and crown, usually higher than 6 meters. Woody plants without a distinct trunk and in a clumped state are dwarf, usually less than 6 meters, and are referred to as shrubs. Plants with herbaceous or fleshy stems with less developed woody parts, whose above-ground parts mostly die in the same year, are called herbaceous plants. The woody species were divided into two sub-sub groups, i.e., trees and shrubs; whereas the herbaceous species were divided into the annual herb and perennial herb. The herbaceous species included 53 annual herbs and 337 perennial herbs. Of these, 53 annual herbaceous plants and 336 perennial herbaceous plants belong to AM plants and only 1 perennial plant belongs to ECM plants. The trees were further divided into deciduous trees and evergreen trees; According to statistics, among the plants in the database, 817 species belong to the AM group, including 153 tree species, 275 shrub species, and 389 herb species; and 167 spe-cies belong to the ECM group, including 160 tree species, 6 shrub species, and 1 herb species. Since the number of shrub species and herb species in ECM plants was too little to compare, for the accuracy of the conclusion, we only compare the leaf δ13C of AM plants and ECM plants in four groups (i.e., total plants, trees, evergreen trees, deciduous trees).
Are the methods adequately described?
( ) ( ) (x) ( )
Response: Accepted. We add a new analysis method (multiple regression analysis) to the Materials Method section and add a detailed description of the method.
Our revised content is as follows:
Lines 141-145: Carbon isotopic value is expressed as the standard notation relative to the Vienna Pee Dee Belemnite standard using the following equation: δ13C=(Rsample/Rstandard-1) × 1,000 (‰), where Rsample and Rstandard are the 13C/12C ratios of the sample and the standard, respectively[32]. In our study, leaf δ13C values of C3 plants were obtained from the database established by Li et al. [7]
Lines 148-152: Then, after having diagnosed the covariance of environmental factors, we attempted to establish the relationship between environmental factors and plant leaf δ13C using multiple regression analysis, comparing the adjusted R2 of the best multiple regression model to assess the relative importance of different environmental factors in deter-mining plant leaf δ13C.
Are the results clearly presented?
( ) ( ) (x) ( )
Response: Accepted. Thank for your Constructive comments. We have carefully revised the results section and added table1 to make the results more accurate. To better present our results, we also added Table1. It’s following:
The analysis of multiple regression analysis was conducted by SPSS software (version 22.0).
Table 1. results of the multiple regression analyses for predicting the leaf δ13C of AM and ECM plants with the combination of all traits
|
Groups |
Mycorrhizal types |
Standardized Coefficients |
R2 |
Sig |
||||||
|
LAT |
LON |
MAP |
MAT |
RH |
Altitude |
SH |
||||
|
Total plants |
AM |
0.182 |
0.120 |
- |
-0.265 |
- |
- |
0.245 |
0.266 |
*** |
|
ECM |
-0.413 |
- |
- |
- |
-0.449 |
0.263 |
- |
0.285 |
*** |
|
|
Trees |
AM |
0.448 |
- |
- |
- |
- |
- |
- |
0.196 |
*** |
|
ECM |
-0.445 |
- |
- |
- |
-0.459 |
0.279 |
- |
0.415 |
*** |
|
|
Evergreen trees |
AM |
0.789 |
-0.417 |
- |
- |
- |
-0.465 |
- |
0.432 |
*** |
|
ECM |
-1.773 |
-1.483 |
- |
- |
-0.583 |
- |
- |
0.494 |
*** |
|
- means that the environmental factors were not included in the best model. *** P<0.001
Add the following:
Lines 180-182: leaf δ13C of ECM plants was significantly higher than that of AM plants (Figure 1c, P<0.001). By contrast, , there was no significant difference in δ13C between AM plants and ECM plants.
Lines 199-201: With the increase of longitude, the decreasing rate of δ13C value of ECM plant leaves was significantly higher than that of AM plant leaves (PAM&ECM<0.001) .
Lines 231-233: In trees, however, the difference between the leaf δ13C maxima of AM plants and ECM plants decreased significantly, and the leaf δ13C maxima of AM plants (-23.26‰) were higher slightly than those of ECM plants (-23.88‰)
Lines 236-237: In evergreen plants, the maximum value of leaf δ13C was -23.88‰ for ECM plants. In deciduous plants, the maximum value of leaf δ13C was -23.46‰ for AM plants.
Lines 247-249: However, the effect of MAP on leaf δ13C of ECM plants was greater than that of AM plants. With the increase in rainfall, the decrease rate of leaf δ13C of ECM plants was twice that of AM plants (Figure 4b).
Lines 282-283: but, the effect of relative humidity change on δ13C in ECM plant leaves was more obvious, and the value of R2 was higher (Figure 6a).
Lines 337-356:
3.4. Stepwise regression analysis of Leaf δ13C and environmental factors in AM and ECM Plants
The model summary was shown in Table 1. When all vegetation types were considered, in AM plants, the value of R2 was 0.266, which indicated that there were 26.6% changes in the response variable (leaf δ13C) because of changes in the combination of four controlled variables including Lat, Lon, MAT, and SH. Among four controlled variables that affected δ13C values, MAT was the most profound environment factor (β=-0.265, P<0.001), while SH was the secondary environment factor (β=0.245, P< 0.001). In ECM plants, the R2 value of the stepwise regression equation was higher than that of AM plants, which was 0.285. According to the results, the variation of leaf δ13C was mainly attributed to four variables, these variables were: RH (β=-0.449, P<0.001), LAT(β=-0.413, P<0.001), Altitude(β=0.263, P<0.001). When only trees were considered, the mul-tivariate stepwise regression equation of leaf δ13C in AM plants only screened out LAT as a sig-nificant influencing factor (β=0.448, P<0.001), It explained 19.6 % of the total variation of leaf δ13C in AM plants. The multivariate stepwise regression equation of δ13C in leaves of ECM plants screened three significant influencing factors, namely Lat (β=-0.445, P<0.001 ), RH (β =-0.459, P<0.001), Altitude (β=0.279, P<0.001), which together explained 41.5 % of the total variation of δ13C in leaves. When only evergreen trees were considered, in AM groups, the best leaf δ13C model showed that Lat(β=0.789, P< 0.001), Lon(β=-0.417, P<0.001), and Altitude(β=-0.465, P<0.001) in combination explained 43.2% of the total variation. Compared to this model, the leaf δ13C of ECM plants was explained by a set of three environmental factors including Lat(β=-1.773, P<0.001), Lon(β=-1.483, P<0.001), and RH(β=-0.583, P<0.001), the explanation rate of the model is 49.4%
Table 1. results of the multiple regression analyses for predicting the leaf δ13C of AM and ECM plants with the combination of all traits
|
Groups |
Mycorrhizal types |
Standardized Coefficients |
R2 |
Sig |
||||||
|
LAT |
LON |
MAP |
MAT |
RH |
Altitude |
SH |
||||
|
Total plants |
AM |
0.182 |
0.120 |
- |
-0.265 |
- |
- |
0.245 |
0.266 |
*** |
|
ECM |
-0.413 |
- |
- |
- |
-0.449 |
0.263 |
- |
0.285 |
*** |
|
|
Trees |
AM |
0.448 |
- |
- |
- |
- |
- |
- |
0.196 |
*** |
|
ECM |
-0.445 |
- |
- |
- |
-0.459 |
0.279 |
- |
0.415 |
*** |
|
|
Evergreen trees |
AM |
0.789 |
-0.417 |
- |
- |
- |
-0.465 |
- |
0.432 |
*** |
|
ECM |
-1.773 |
-1.483 |
- |
- |
-0.583 |
- |
- |
0.494 |
*** |
|
- means that the environmental factors were not included in the best model. *** P<0.001
Are the conclusions supported by the results?
( ) ( ) (x) ( )
Response: Accepted. We have modified the conclusion section to make the conclusion more complete.
The main content of the conclusion:
- The variations of leaf δ13C in AM and ECM plants are different under differernt life forms
- The effects of climateon leaf δ13C are closely related to mycorrhizal types
- The leaf δ13C of AM plants was mainly affected by temperature, while the leaf δ13C of ECM plants was more sensitive to moisture content.
If there are any problems, we will make further modification.
Comments and Suggestions for Authors
Point 1: Line 88: Please explain why it is important to concentrate on the C3 plant for the present study
Response 1: Carbon dioxide (CO2) in the atmosphere is sequestered into plants by photosynthesis, during CO2 fixation by plants, the carbon isotopes (13C and 12C) are discriminated, and this discrimination results in different 13C abundances (δ13C) in different plant species. In the photosynthesis process of plants, the isotopic fractionation ability of C3 plants is better than that of C4 plants with good recognition. And most of C3 plants are positive plants in land plants, and the δ13C of C3 plants can better help us understand the relationship between land plants and their environment.
Point 2: To provide further clarification, details and active database links can be provided.
Response 2: Thank for your reminding. We have uploaded the data. The raw data for this study available via https://figshare.com/s/fcfdd37541c8ba1b8278.
Point 3: Line 93-94: It is necessary to provide a thorough explanation of the procedure utilized to identify the particular plant species.
Response 3: Accepted. We have described in detail the procedures used for the identification of plant species in the database.
Line 96-102: Moreover, in this study, we call trees those woody plants with independent trunks that occur from the roots, with distinct trunks and crowns and usually higher than six meters. Woody plants without a distinct trunk and in a clumped state are dwarfed and usually below six meters and are called shrubs. Plants with herbaceous or fleshy stems with less developed woody parts, and whose above-ground parts mostly wither in the same year, are called herbs.
Point 4: Data of annual herbs and the perennial herb is missing. Please elaborate.
Response 4: Accept. Thanks for the reviewer's constructive suggestions. We have supplemented the data for annual and perennial herbs and listed the number of annual and perennial herbs in AM plants and ECM plants.
Line 107-109: The herbaceous species included 56 annual herbs and 360 perennial herbs. Of these, 56 annual herbaceous plants and 359 perennial herbaceous plants belong to AM plants and only one perennial plant belongs to ECM plants.
Point 5: Lines 114–115: Please describe the steps taken to estimate the leaf δ13C.
Response 5: Thank the expert for her/his approval. We have added the steps taken to estimate leaf δ13C in the materials and methods section. Plant leaf δ13C is usually measured by a mass spectrometer and an elemental analyzer in a continuous flow mode. Carbon isotopic value is expressed as the standard notation relative to the Vienna Pee Dee Belemnite standard using the following equation: δ13C =(Rsample/Rstandard - 1) × 1,000 (‰), where Rsample and Rstandard are the 13C/12C ratios of the sample and the standard, respectively. In this study, the leaf δ13C trait data in China were obtained from the database established by Li et al. (2017).
Line 123-127: Carbon isotopic value is expressed as the standard notation relative to the Vienna Pee Dee Belemnite standard using the following equation: δ13C =(Rsample/Rstandard-1) × 1,000 (‰), where Rsample and Rstandard are the 13C /12C ratios of the sample and the standard, respectively. In our study, leaf δ13C values of C3 plants were obtained from the database established by Li et al. [19]
Point 6: Analysis of leaf δ13C in AM plants and ECM plants for herbs is missing (Figure 1 to Figure 8).
Response 6: Our objective was to compare the variability in the relationship between leaf δ13C and the environment among different mycorrhizal types of plants, and in our database, AM plants cover a larger number of plant life types, including evergreen trees, deciduous trees, annual herbs, and perennial herbs, while ectomycorrhizae are mainly dominated by trees and largely exclude shrubs and graminoids (in our database, ECM plants have only 6 shrubs and only 1 for herbaceous in our database). Therefore, considering the accuracy of the results, we explored the effects of AM and ECM on terrestrial ecosystems by comparing trees with different mycorrhizal types.
Point 7: Define the significance of R2 with respect to the present study and maintain consistency when presenting it, as in Figure 2, where it is referred to as the R2, and in lines 183, 184.
Response 7: R2 in this study represents the ability of environmental factors ( such as latitude, longitude, temperature, precipitation, etc. ) to explain leaf δ13C. In addition, I am very grateful to the experts for pointing out my mistakes in expression. We have carefully checked the full text and standardized the expression of R2.
Thank you for your comments on our manuscript again. We have tried our best to modify the manuscript. We will make further modification to improve the quality of the manuscript, If there are any problems.

Round 2
Reviewer 1 Report
The new version of the manuscript is now acceptable as a scientific work.
Reviewer 2 Report
The authors have responded well to my suggestions and manuscript may be accepted.